# Weyl magnons in breathing pyrochlore antiferromagnets

Fei-Ye Li[1,*], Yao-Dong Li[2,*], Yong Baek Kim[3,4], Leon Balents[5], Yue Yu[6,7,8] & Gang Chen[6,7,8]

Frustrated quantum magnets not only provide exotic ground states and unusual magnetic structures, but also support unconventional excitations in many cases. Using a physically relevant spin model for a breathing pyrochlore lattice, we discuss the presence of topological linear band crossings of magnons in antiferromagnets. These are the analogues of Weyl fermions in electronic systems, which we dub Weyl magnons. The bulk Weyl magnon implies the presence of chiral magnon surface states forming arcs at finite energy. We argue that such antiferromagnets present a unique example, in which Weyl points can be manipulated in situ in the laboratory by applied fields. We discuss their appearance specifically in the breathing pyrochlore lattice, and give some general discussion of conditions to find Weyl magnons, and how they may be probed experimentally. Our work may inspire a re-examination of the magnetic excitations in many magnetically ordered systems.

[1] Institute of Theoretical Physics, Chinese Academy of Sciences, Beijing 100190, People's Republic of China. [2] School of Computer Science, Fudan University, Shanghai 200433, People's Republic of China. [3] Department of Physics, University of Toronto, Canadian Institute for Advanced Research, Quantum Materials Program, Toronto, Ontario, Canada MSG1Z8. [4] School of Physics, Korea Institute for Advanced Study, Seoul 130-722, Korea. [5] Kavli Institute for Theoretical Physics, Santa Barbara, California 93106, USA. [6] State Key Laboratory of Surface Physics and Department of Physics, Fudan University, Shanghai 200433, People's Republic of China. [7] Center for Field Theory and Particle Physics, Department of Physics, Fudan University, Shanghai 200433, People's Republic of China. [8] Collaborative Innovation Center of Advanced Microstructures, Nanjing 210093, People's Republic of China. * These authors contributed equally to this work. Correspondence and requests for materials should be addressed to G.C. (email: gangchen.physics@gmail.com).

It is commonly thought that the spin ordering pattern of a magnetic insulator uniquely specifies the state of the system[1], and indeed the ground state of such materials is usually well-described by a simple product state of little fundamental interest. However, in view of recent developments in the study of topological properties of periodic media[2,3], it is possible that even such a product-like ground state can support topologically non-trivial excited state band structure. Topological properties of bands have been studied previously for electrons in solids governed by Schrödinger's equations[2,3], for photons in dielectric superlattices governed by Maxwell's equations[4,5], for phonons governed by Newton's equations[4], and even for fractionalized spinon excitation in spin liquids[6,7]. Here we apply these ideas to magnons governed by the equations for spin waves in an ordered antiferromagnet. We consider a concrete magnetic system, namely, the Cr-based breathing pyrochlore, and explicitly demonstrate that it supports Weyl magnon excitations with a linear band touching in the spin-wave spectrum of the magnetic ordered phase. The Weyl magnon is analogous to a Weyl fermion[8–11] in electronic systems, but has bosonic rather than fermionic statistics, similar to Weyl points in photonic systems[5]. In contrast to the other three categories of systems, the band structure of magnons in antiferromagnets is highly tunable *in situ* by application of readily available magnetic fields, which is a consequence of the spontaneous symmetry breaking of the antiferromagnet ground state and the relatively low-energy scale for magnetic interactions in most solids. Thus one can envision moving, creating and annihilating Weyl points in the laboratory in a single experiment.

To explore Weyl magnons, we focus on a concrete and physical model system, the breathing pyrochlore antiferromagnet. This is a generalization of the common pyrochlore structure, which consists of a network of corner sharing tetrahedra, with magnetic ions at the corners. In the breathing pyrochlore, alternate tetrahedra are uniformly expanded and contracted in size[12–16]. As a result, the structure lacks an inversion center, and in general up-pointing and down-pointing tetrahedral units are inequivalent. We consider below a spin model for the breathing pyrochlore, which generalizes and includes the uniform limit, and displays Weyl points even in the uniform case. We obtain the full phase diagram of this spin model and the magnetic excitations in different phases. The experimental consequences of Weyl magnons and the general conditions for their occurrence in spin systems are predicted and discussed.

## Results

**Spin model**. We consider $Cr^{3+}$ ions in the breathing pyrochlore lattice. There are several compounds with this structure, including $LiGaCr_4O_8$ and $LiInCr_4O_8$, which have been recently studied[13,14]. In this $3d^3$ electron configuration the orbital angular momentum is fully quenched and the local moment is well-described by the isotropic Heisenberg exchange and a total spin $S = 3/2$ according to Hund's rules. The minimal spin model is given as

$$H = J \sum_{\langle ij \rangle \in \mathrm{u}} \mathbf{S}_i \cdot \mathbf{S}_j + J' \sum_{\langle ij \rangle \in \mathrm{d}} \mathbf{S}_i \cdot \mathbf{S}_j + D \sum_i (\mathbf{S}_i \cdot \hat{\mathbf{z}}_i)^2, \quad (1)$$

Since spin-orbit coupling is weak, the interaction between the local moments is primarily where we have supplemented the Heisenberg model with a local spin anisotropy[17], which is generically allowed by the $D_{3d}$ point group symmetry at the Cr site. The anisotropic direction $\hat{\mathbf{z}}_i$ is the local [111] direction that points into the center of each tetrahedron and is specified for each sublattice (Methods). Here $J$ and $J'$ are the exchange couplings between the nearest-neighbour spins on the up-pointing and

down-pointing tetrahedra (Fig. 1), respectively. The large and negative Curie–Weiss temperatures of the Cr-based breathing pyrochlores indicate the strong atomic force microscopy interactions, hence we take $J > 0$, $J' > 0$. Because the up-pointing and down-pointing tetrahedra have different sizes, one thus expects $J \neq J'$. In this work, however, we will study this model in a general parameter setting. The atomic force microscopy exchange interactions favour zero total spin on each up-pointing (down-pointing) tetrahedron, that is, $\sum_{i \in \mathrm{u}} \mathbf{S}_i = 0$ ($\sum_{i \in \mathrm{d}} \mathbf{S}_i = 0$). As for the regular pyrochlore lattice[18], the classical ground state of the exchange part of the Hamiltonian is extensively degenerate.

**Ground states and quantum order by disorder**. We first consider easy-axis spin anisotropy with $D < 0$. This favours the spin to be aligned with its local [111] axis. It turns out that this condition can be satisfied while simultaneously optimizing the exchange interaction. This gives a unique classical ground state (up to a 2-fold degeneracy from the time-reversal operation) that has an all-in all-out magnetic order. The magnetic excitation of this ordered state is fully gapped and the energy gap ($\Delta$) is simply set by the easy-axis spin anisotropy with $\Delta = 3|D|$ (Methods).

With the easy-plane anisotropy, $D > 0$, the spin prefers to orient in the $xy$ plane of the local coordinate system at each sublattice. This requirement can also be satisfied while simultaneously optimizing the exchange. Moreover, there exists an accidental $U(1)$ degeneracy of the classical ground state that we parametrize as

$$\mathbf{S}_i^{\mathrm{cl}} \equiv S\hat{\mathbf{m}}_i = S(\cos\theta\,\hat{\mathbf{x}}_i + \sin\theta\,\hat{\mathbf{y}}_i), \quad (2)$$

where $\hat{\mathbf{x}}_i$ ($\hat{\mathbf{y}}_i$) is the unit vector along the local $x$ ($y$) axis in the local coordinate system at site $i$ (Methods), the unit vector $\hat{\mathbf{m}}_i$

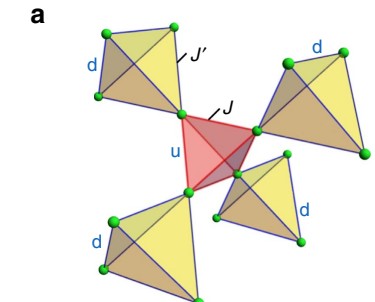

**a**

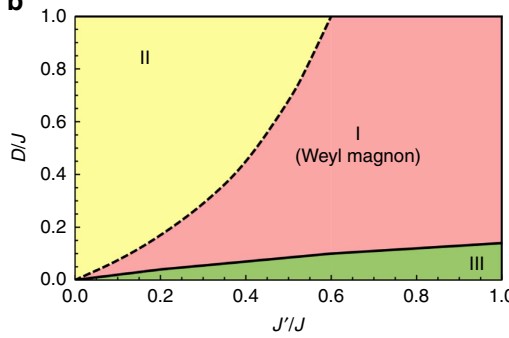

**b**

**Figure 1 | The breathing pyrochlore and the phase diagram.**
(**a**) The breathing pyrochlore. The letter u(d) refers to the up-pointing (down-pointing) tetrahedra and $J(J')$ indicates the nearest-neighbour exchange couplings on the up-pointing (down-pointing) tetrahedra. (**b**) The phase diagram. Regions I and II have the same magnetic order and belong to the same phase, but the magnetic excitations of the two regions are topologically distinct. Region III has a different magnetic order. The details of the phase diagram are discussed in the main text.

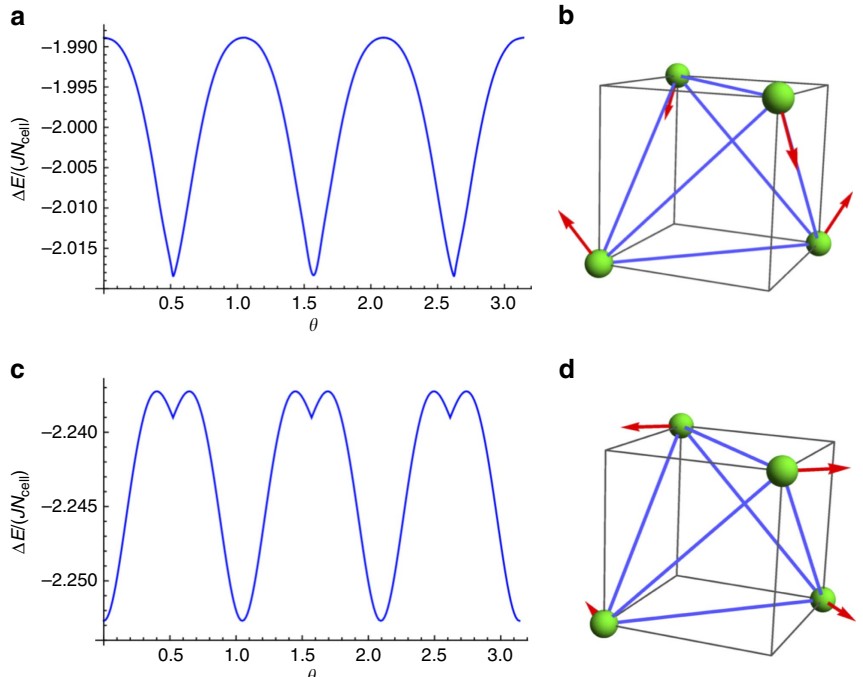

**Figure 2 | Quantum zero point energy and the magnetic order.** We have chosen the representative parameters in regions I and III with $D = 0.2J$, $J' = 0.6J$ in (**a**) and $D = 0.05J$, $J' = 0.6J$ in (**c**), respectively. (**b**) The magnetic order in regions I and II with $\theta = \pi/2$ and the spins pointing along the local $\hat{\mathbf{y}}$. (**d**) The magnetic order in region III with $\theta = 0$ and the spins pointing along the local $\hat{\mathbf{x}}$.

points in the local $xy$ plane, and the angular variable $\theta$ captures the $U(1)$ degeneracy. This is the same form of degeneracy found for the $S = 1/2$ pyrochlore $Er_2Ti_2O_7$ in ref. 19, where it was noted that the degeneracy is accidental, that is, not protected by any symmetry, and hence will be lifted by quantum fluctuations. The same holds for the breathing pyrochlore, as we show now using linear spin-wave theory. We introduce the Holstein–Primakoff bosons to express the spin operators as $\mathbf{S}_i \cdot \hat{\mathbf{m}}_i = S - a_i^\dagger a_i$, $\mathbf{S}_i \cdot \hat{\mathbf{z}}_i = (2S)^{1/2}(a_i + a_i^\dagger)/2$, and $\mathbf{S}_i \cdot (\hat{\mathbf{m}}_i \times \hat{\mathbf{z}}_i) = (2S)^{1/2}(a_i - a_i^\dagger)/(2i)$. Keeping terms in the spin Hamiltonian $H$ up to the quadratic order in the Holstein–Primakoff bosons, one can readily write down the spin-wave Hamiltonian as

$$
\begin{aligned}
H_{\text{sw}} = \; & \sum_{\mathbf{k}} \sum_{\mu,\nu} \Big[ A_{\mu\nu}(\mathbf{k}) a_{k,\mu}^\dagger a_{k,\nu} + B_{\mu\nu}(\mathbf{k}) a_{-k,\mu} a_{k,\nu} \\
& + B_{\mu\nu}^*(-\mathbf{k}) a_{k,\mu}^\dagger a_{-k,\nu}^\dagger \Big] + E_{\text{cl}},
\end{aligned}
\tag{3}
$$

where $E_{\text{cl}}$ is the classical ground state energy, and $A_{\mu\nu}$, $B_{\mu\nu}$ satisfy $A_{\mu\nu}(\mathbf{k}) = A_{\nu\mu}^*(\mathbf{k})$, $B_{\mu\nu}(\mathbf{k}) = B_{\nu\mu}(-\mathbf{k})$ and depend on the angular variable $\theta$. Although the classical energy $E_{\text{cl}}$ is independent of $\theta$ due to the $U(1)$ degeneracy, the quantum zero point energy $\Delta E$ of the spin-wave modes depends on $\theta$, and is given by $\Delta E = \sum_{\mathbf{k}} \sum_{\mu} \frac{1}{2}[\omega_\mu(\mathbf{k}) - A_{\mu\mu}(\mathbf{k})]$, where $\omega_\mu(\mathbf{k})$ is the excitation energy of the $\mu$-th spin-wave mode at momentum $\mathbf{k}$ and is determined for every classical spin ground state. The minimum of $\Delta E$ occurs at $\theta = \pi/6 + n\pi/3$ $(n\pi/3)$ with $n \in \mathbb{Z}$ in regions I and II (region III). The discrete minima and the corresponding magnetic orders are equivalent under space group symmetry operations. The $U(1)$ degeneracy of the classical ground states is thus broken by quantum fluctuations. This is the well-known phenomenon known as quantum order by disorder[19–22]. The resulting optimal state is a non-collinear one in which each spin points along its local [112] ([1$\bar{1}$0]) lattice direction in regions I and II (region III), see Fig. 2.

To obtain the phase diagram in Fig. 1, we have implemented the semiclassical approach and included the quantum fluctuation within linear spin-wave theory. This treatment may underestimate the quantum fluctuation in the parameter regimes when $J \gg J'$, $D$ or $J' \gg J$, $D$. In the latter regimes, one may first consider the tetrahedron with the strongest coupling and treat other couplings as perturbations. The ground state in these regimes is likely to be non-magnetic and will be addressed in the future work. For the purpose of the current work, we will focus on the ordered ground states in Figs 1 and 2.

**Magnon Weyl nodes and surface states.** Regions I and II have the same magnetically ordered structure with the same order parameter and belong to the same phase. Although the ground states are characterized by the same order parameter, the magnetic excitations of the two regions are topologically distinct. The magnetic excitation in region I has Weyl band touchings, while the region II does not. To further clarify this, we choose $\theta = \pi/2$ and thus fix the magnetic order to orient along the $\hat{\mathbf{y}}$ directions of the local coordinate systems. Using linear spin-wave theory, we obtain the magnetic excitation spectrum with respect to this magnetic state for regions I and II. In Fig. 3a, we depict a representative excitation spectrum along the high-symmetry lines in the Brillouin zone for region I.

Two qualitative features are clear in the magnon spectrum of Fig. 3a. First, we observe a gapless mode at the $\Gamma$ point. This pseudo-Goldstone mode is an artifact of the linear spin-wave approximation, and a small gap is expected to be generated by anharmonic effects[19]. Secondly, the spectrum in Fig. 3a has a linear band touching at a point along the line between $\Gamma$ and $X$. In fact, as we show in Fig. 3b, there are in total four such linear band touchings. The bands separate linearly in all directions away from these touchings, which are thus Weyl nodes in the magnon spectrum. Just like Weyl nodes of non-degenerate electron bands[8], the magnon Weyl points are sources and sinks of Berry curvature and are characterized by a discrete chirality taking

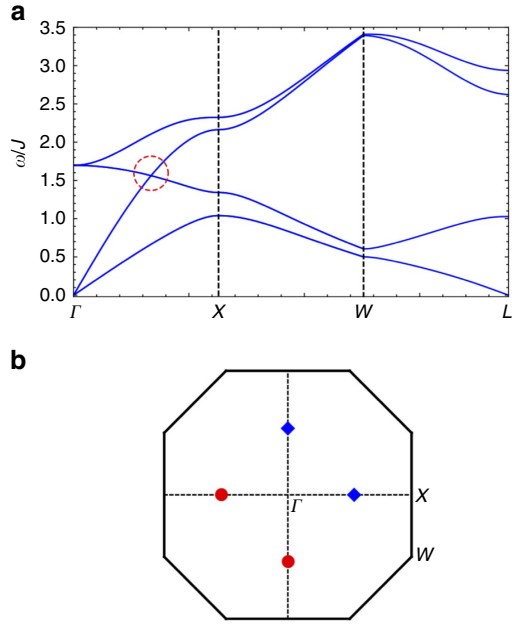

**Figure 3 | The representative spin-wave spectrum and the Weyl nodes of region I.** (**a**) The spin-wave spectrum along high-symmetry momentum lines with a linear band touching that is marked with a (red) dashed circle. (**b**) Four Weyl nodes are located at ($\pm k_0$, 0, 0), (0, $\pm k_0$, 0) with $k_0 = 1.072\pi$ in the $xy$ plane of the Brillouin zone. The (red) circle has an opposite chirality from the (blue) diamond. In the figure, we have set $D = 0.2J$, $J' = 0.6J$ and $\theta = \pi/2$.

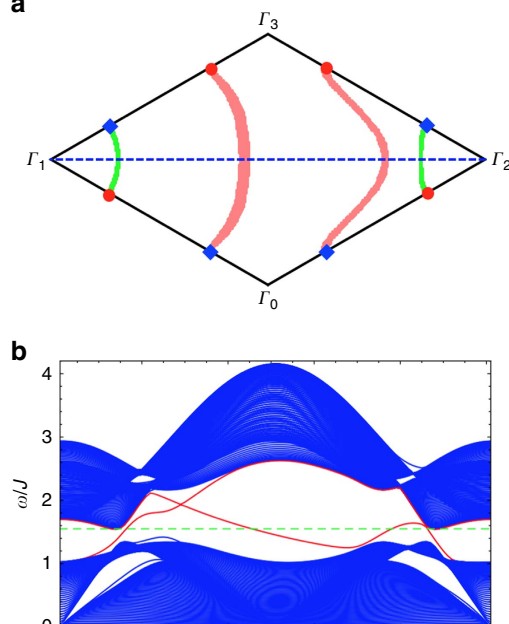

**Figure 4 | Surface states of a slab.** The slab is cleaved along the [11$\bar{1}$] surface, setting $D = 0.2J$, $J' = 0.6J$ and $\theta = \pi/2$. (**a**) Magnon arcs in the surface Brillouin zone. $\Gamma_0$ is the origin of the Brillouin zone and two reciprocal lattice vectors are $\overrightarrow{\Gamma_0\Gamma_1} = \frac{4\pi}{3}[2\bar{1}1]$, $\overrightarrow{\Gamma_0\Gamma_2} = \frac{4\pi}{3}[\bar{1}21]$. The surface states with $E = E_{Weyl}$ form arcs connecting the Weyl nodes with different chiralities, where $E_{Weyl}$ is the energy of the bulk Weyl nodes. States along the two pink longer (green shorter) arcs are localized in the top (bottom) surface. (**b**) The (blue) bulk magnon excitations and the (red) chiral surface states along $\overrightarrow{\Gamma_1\Gamma_2}$. The (green) dashed line indicates $E = E_{Weyl}$.

values $\pm 1$. Unlike in an electronic Weyl semimetal, where one can tune the Fermi energy to the Weyl nodes by varying the electron density, the magnon Weyl nodes must necessarily appear at finite energies because of the bosonic nature of magnons.

Due to the bulk-edge correspondence, we expect magnon arc states bound to any surface which possesses non-trivial projections of the bulk Weyl points. This is indeed observed in Fig. 4. The chiral magnon arcs appear at non-zero energy and connect the bulk magnon Weyl nodes with opposite chiralities, as expected.

Once the magnon Weyl nodes emerge in the magnon spectrum, they are topologically robust and exist over a finite regime in the parameter space. We find that the magnon Weyl nodes exist in region I. As the couplings are varied so that the boundary with region II is approached, the magnon Weyl nodes move together and annihilate in pairs when the boundary is reached. In region II, there is no such (Weyl) band crossing, qualitatively distinguishing region II from region I.

**Manipulating Weyl nodes by external magnetic fields.** When we apply an external magnetic field to the system, the spin only couples to the field via a Zeeman coupling. This is quite different from the case of electronic systems, in which a magnetic field also has an orbital effect, which leads to cyclotron motion of electrons and a transformation from ordinary bands into Landau ones. In the latter case, the meaning of quasi-momentum is irrevocably changed by an applied field, and one cannot follow the Weyl point evolution with field. By contrast, since magnons are neutral, there is no orbital effect, and quasi-momentum and the Weyl points themselves remain well-defined even for strong fields. Therefore, a magnetic field can be used to manipulate the Weyl nodes. To demonstrate this explicitly, we focus on one specific classical order in region I and apply a magnetic field along the global $z$ direction. The magnetic field perturbs the classical

ground state and indirectly changes the spin-wave Hamiltonian. As we show in Fig. 5, the Weyl nodes are shifted gradually and finally annihilated when the magnetic field is increased.

## Discussion

We have explicitly shown the presence of Weyl nodes in a simple and physically relevant model for the breathing pyrochlore lattice antiferromagnet. Weyl points may also be present in other pyrochlores for which the exchange is more complicated. The spin-wave spectra of the highly anisotropic spin-1/2 pyrochlores $Yb_2Ti_2O_7$ and $Er_2Ti_2O_7$ have been extensively studied[19,23]. Re-examined here in the light of topology, we see that they are present already in the spin-wave spectra of $Yb_2Ti_2O_7$ and $Er_2Ti_2O_7$ in the external magnetic fields. Thus we think that Weyl points can be present in many magnetic materials of current interest.

Beyond these specific examples, we may ask what are the conditions necessary to find Weyl points in the magnon spectrum? In electronic systems, these points are symmetry prevented, meaning that if both inversion $\mathcal{P}$ and time-reversal symmetry $\mathcal{T}$ are present, Weyl points cannot occur. This is because in that case, a two-fold Kramers' degeneracy of bands occurs, and any crossing must involve two and not four bands. For magnons, there is never a Kramer's degeneracy. This is because magnons are integer spin excitations (even when the spin is not a good quantum number they are superpositions of integer spin excitations), which do not obey Kramer's theorem because $\mathcal{T}^2 = +1$ in this case. Moreover, in general the magnetic order which underlies magnons already breaks time-reversal symmetry. This suggests that Weyl points may be generically allowed.

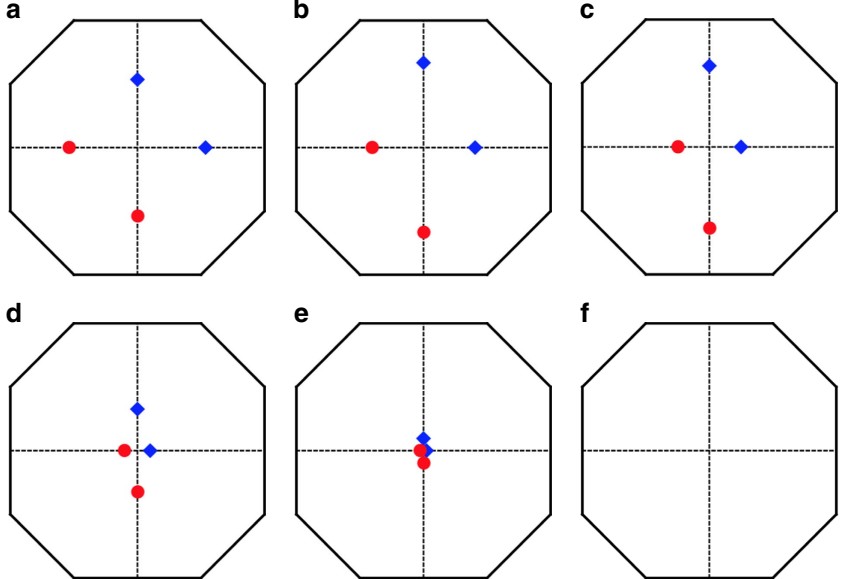

**Figure 5 | The evolution of Weyl nodes under the magnetic field.** Applying a magnetic field along the global $z$ direction, $\mathbf{B} = B\hat{\mathbf{z}}$, Weyl nodes are shifted but still in $k_z = 0$ plane. They are annihilated at $\Gamma$ when magnetic field is strong enough. Red and blue indicate the opposite chirality. (**a,f**): $B = 0$, $0.1J$, $0.5J$, $0.9J$, $1.0J$, $1.1J$. We have set $D = 0.2J$, $J' = 0.6J$ and $\theta = \pi/2$.

However, there are some conditions under which Weyl points are prohibited. In particular, many magnetically ordered systems possess not time-reversal but a complex conjugation symmetry $\mathcal{C}$. This is the case for any Heisenberg Hamiltonian with a collinear ordered ground state, but it can occur more generally. If, in addition, the system possesses inversion symmetry $\mathcal{P}$, then Weyl points are prohibited. This can be understood from the Berry curvature[24,25], $\Omega_\mu(\mathbf{k}) = i\epsilon_{\mu\nu\lambda}\langle\partial_\nu\mathbf{k}|\partial_\lambda\mathbf{k}\rangle$, defined in terms of the exact magnon eigenstates $|\mathbf{k}\rangle$ of a given magnon band. The Berry curvature is an effective magnetic field in momentum space, and a Weyl point is defined as a delta-function source (divergence) of this curvature. If $\mathcal{P}$ is valid, one has $\Omega_\mu(\mathbf{k}) = \Omega_\mu(-\mathbf{k})$, while $\mathcal{C}$ implies $\Omega_\mu(\mathbf{k}) = -\Omega_\mu(-\mathbf{k})$. Hence the combination requires $\Omega_\mu(\mathbf{k}) = 0$, prohibiting any Berry curvature at all, and also obviously Weyl points.

This shows that in the simplest magnetically ordered systems, Weyl points are not allowed. There may be other conditions prohibiting Weyl points, or constraining them. A trivial condition is that one needs at least two magnon bands to form Weyl points, which prohibits them in some simple ferromagnets. In the case studied in this paper a two-fold rotation axis locks the Weyl points along the $\Gamma$–$X$ axes. A full treatment of the necessary and sufficient conditions for Weyl points may be part of a topological spin-wave theory[26,27], to be developed in the future.

Now we turn to experimental implications. The most natural probe of the bulk magnon Weyl nodes as well as the surface magnon arc states is inelastic neutron scattering. Because of the surface dependence of the magnon arc states, one could study the system with different slab geometries and surface orientations. For example, for the $[11\bar{1}]$ surfaces, one would observe two disconnected arcs on both up and down surfaces (Fig. 4). In contrast, one would observe two loops across the surface Brillouin zone for the $[110]$ surfaces because two pairs of Weyl nodes with different chiralities are projected onto the same points (Methods).

The Weyl magnon can be potentially detected optically. Close to the Weyl nodes, a vertical transition can occur with arbitrarily small energy. Because the lower state is empty at zero temperature in equilibrium, it may be beneficial to use a pump-probe approach to measure the optical absorption. Then one may be able to observe optical absorption at low frequency[28], when the

**Table 1 | The local axis for the four sublattices of the breathing pyrochlore lattice.**

| $\mu$ | $\hat{\mathbf{x}}_\mu$ | $\hat{\mathbf{y}}_\mu$ | $\hat{\mathbf{z}}_\mu$ |
|---|---|---|---|
| 1 | $\frac{1}{\sqrt{2}}[\bar{1}10]$ | $\frac{1}{\sqrt{6}}[\bar{1}\bar{1}2]$ | $\frac{1}{\sqrt{3}}[111]$ |
| 2 | $\frac{1}{\sqrt{2}}[\bar{1}\bar{1}0]$ | $\frac{1}{\sqrt{6}}[\bar{1}1\bar{2}]$ | $\frac{1}{\sqrt{3}}[11\bar{1}]$ |
| 3 | $\frac{1}{\sqrt{2}}[110]$ | $\frac{1}{\sqrt{6}}[1\bar{1}\bar{2}]$ | $\frac{1}{\sqrt{3}}[\bar{1}1\bar{1}]$ |
| 4 | $\frac{1}{\sqrt{2}}[1\bar{1}0]$ | $\frac{1}{\sqrt{6}}[112]$ | $\frac{1}{\sqrt{3}}[\bar{1}\bar{1}1]$ |

The letter $\mu$ refers to the sublattice, and $(\hat{\mathbf{x}}_\mu, \hat{\mathbf{y}}_\mu, \hat{\mathbf{z}}_\mu)$ defines the local coordinate system at the $\mu$-th sublattice.

lower magnon bands have enough population. In addition to the spectroscopic property, the presence of the Weyl magnon spectrum may lead to a thermal Hall effect, just like the Weyl fermion that gives rise to the anomalous Hall current in electronic systems[29,30]. Furthermore, one could use magnetic field to control thermal Hall signal[31–33] despite the absence of the Lorentz coupling of the spin to the external magnetic field. Again due to population effects, the thermal Hall signal from Weyl magnons will be suppressed at low temperature, but could be enhanced by optical pumping.

Although the existing experiments suggest that both $LiGaCr_4O_8$ and $LiInCr_4O_8$ develop the antiferromagnetic long-range orders at low temperature[13,14], the precise structures of the magnetic order in these two systems are not yet clear at this stage. Therefore, it is certainly of interest to confirm the magnetic order and detect possible Weyl magnon excitations in these systems and other three dimensional Mott insulators with long-range magnetic orders.

To summarize, we have studied a realistic spin model on the Cr-based breathing pyrochlore lattice. We show that the combination of the single-ion spin anisotropy and the superexchange interaction leads to novel magnetically ordered ground states. Remarkably, the magnetic excitations in a large parameter regime develops magnon Weyl nodes in the magnon spectrum. We expect that Weyl magnons may exist broadly in many ordered magnets. We propose a number of experiments that can test the presence of the Weyl magnons.

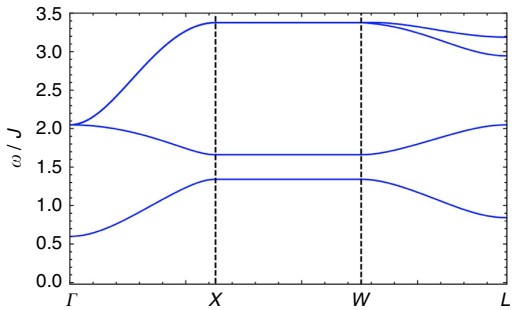

**Figure 6 | The spin-wave spectrum of the all-in all-out state.** In this Figure, we set $D = -0.2J$, $J' = 0.6J$.

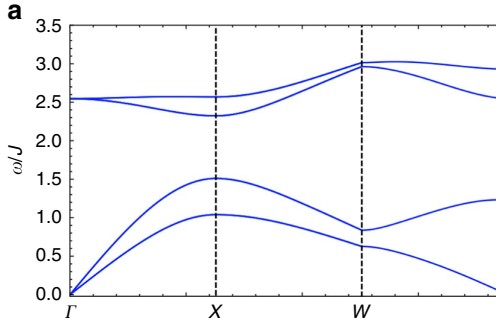

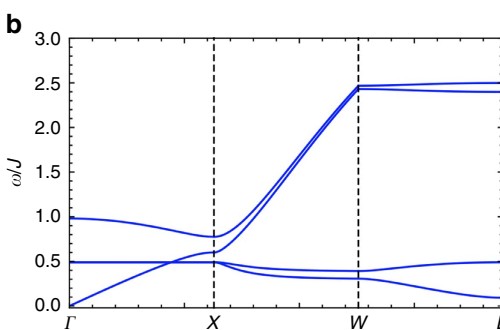

**Figure 7 | The spin-wave spectrum of representative points in regions II and III.** In the figure, we have chosen the parameters as (**a**) $D = 0.6J$, $J' = 0.2J$, $\theta = \pi/2$ and (**b**) $D = 0.05J$, $J' = 0.6J$, $\theta = \pi/3$.

## Methods

**Local coordinate system.** The local coordinate system is defined for each sublattice and is given in Table 1.

**Spin-wave spectrum for the all-in all-out state.** For the easy-axis anisotropy with $D < 0$, we have all-in all-out magnetic order, and the spin-wave $H_{sw}$ in equation (3) is specified by the entries,

$$A_{\mu\mu}(\mathbf{k}) = S(-2D + J + J'), \qquad (4)$$

$$A_{\mu\nu}(\mathbf{k}) = -\frac{1}{3} S J_{\mu\nu}, \qquad (5)$$

$$B_{\mu\mu}(\mathbf{k}) = 0, \qquad (6)$$

$$B_{\mu\nu}(\mathbf{k}) = \frac{1}{3} S J_{\mu\nu} e^{i\phi_{\mu\nu}}, \qquad (7)$$

where $J_{\mu\nu} = J + J' e^{-i\mathbf{k}\cdot(\mathbf{b}_\nu - \mathbf{b}_\mu)}$ ($\mu \neq \nu$), $\mathbf{b}_1 = [000]$, $\mathbf{b}_2 = 1/2[011]$, $\mathbf{b}_3 = 1/2[101]$, $\mathbf{b}_4 = 1/2[110]$ and $\phi_{\mu\nu} = \phi_{\nu\mu}$ ($\mu \neq \nu$), with

$$\phi_{12} = \phi_{34} = -\frac{\pi}{3}, \phi_{13} = \phi_{24} = \frac{\pi}{3}, \phi_{14} = \phi_{23} = \pi. \qquad (8)$$

The magnetic excitation of this ordered state is fully gapped and the energy gap ($\Delta$) is simply set by the easy-axis spin anisotropy with $\Delta = 3|D|$ (Fig. 6).



**Figure 8 | Surface arc states of a slab.** The slab is cleaved along the [110] surface, setting $D = 0.2J$, $J' = 0.6J$ and $\theta = \pi/2$. $\Gamma_0$ is the origin of the surface Brillouin zone, and two reciprocal lattice vectors are $\overrightarrow{\Gamma_0\Gamma_1} = 2\pi[1\bar{1}0]$, $\overrightarrow{\Gamma_0\Gamma_2} = 2\pi[001]$. The Surface states with $E = E_{Weyl}$ form arcs connecting the projections of Weyl nodes, where $E_{Weyl}$ is the energy of the bulk Weyl nodes. Note each pair of nodes are projected to the same position. Pink (Green) arcs are localized in one (the other) surface.

**In-plane ordered states.** For the in-plane magnetic orders, the entries of $H_{sw}$ in equation (3) are given by

$$A_{\mu\mu}(\mathbf{k}) = S(D + J + J'), \qquad (9)$$

$$A_{\mu\nu}(\mathbf{k}) = -\frac{1}{3} S J_{\mu\nu} \left(1 + \cos\left(2\theta + \phi_{\mu\nu}\right)\right), \qquad (10)$$

$$B_{\mu\mu}(\mathbf{k}) = \frac{1}{2} S D, \qquad (11)$$

$$B_{\mu\nu}(\mathbf{k}) = \frac{1}{6} S J_{\mu\nu} \Big[ \cos\left(2\theta + \phi_{\mu\nu}\right) - i2\sqrt{2} \cos\left(\theta - \phi_{\mu\nu}\right) \Big], \qquad (12)$$

where $J_{\mu\nu}$ and $\phi_{\mu\nu}$ are the same as the ones that are defined for the all-in all-out state. In Fig. 7, we plot the spin-wave spectrum for regions II and III. For region III, there exists a band crossing between a dispersive band and two (degenerate) flat bands from $\Gamma$ to $X$. This band crossing may turn into Weyl band touchings if one includes extra spin interactions that make the flat bands non-degenerate and dispersive.

Finally in Fig. 8, we depict the surface arcs for the [110] surfaces in region II. For this surface, each pair of nodes are projected to the same position, and the surface arcs form two loops across the surface Brillouin zone and connect the Weyl nodes.

**Data availability.** The data that support the findings of this study are available from the corresponding author (G.C.) upon request.

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

## Acknowledgements

We thank Xi Dai, Yang Qi, R. Shindou, Nanlin Wang, Zhong Wang, Xincheng Xie, Fan Zhang, Fuchun Zhang and Yi Zhou for discussion and comments. This work is supported by the Start-Up Fund of Fudan University and the Thousand-Youth-Talent Program of People's Republic of China (F.-Y.L., Y.D.L., G.C.), the NSERC (Y.B.K.), the DOE Office of Basic Energy Sciences DE-FG02-08ER46524 (L.B.), and the 973 Program of MOST of China 2012CB821402, NNSF of China 11174298, 11474061 (Y.Y.).

## Author contributions

G.C. designed this project. F.-Y.L., Y.-D.L. and G.C. performed the calculation. F.-Y.L., L.B. and G.C. wrote the manuscript. All authors commented on the manuscript and the results.

## Additional information

**Competing financial interests:** The authors declare no competing financial interests.

