## [Peer review file · Nature Communications]

Reviewers' comments:

Reviewer #1 (Remarks to the Author):

This is an interesting paper about the emergence of topological linear band crossing in the three-dimensional magnonic excitation spectrum, which is dubbed as "Weyl magnons", in analog to "Weyl semi-metals" discovered in electronic systems recently. Based on a physically relevant spin model and the standard spin density wave theory (Holstein-Primarkoff bosons), the authors show that such type of magnonic excitations can be found in the Cr-based breathing pyrochlore lattice. The result is sound and the calculation is standard. In my mind, this paper is possible to arouse research interests of people from the fields of both topological physics and magnetic materials and thus potentially has an impact for both fields. Before I can make a recommendation, I would like to bring the following questions to the authors.

1. As discussed in the paper, one worry to observe "Weyl magnons" in experiments is the population effect since the Weyl point in magnonic excitations lies in a relative high energy. This makes the current work less interesting and also difficult for experiments. Could the authors give an estimate of temperature range to observe such type of magnonic excitations in realistic materials? Is the energy range for magnonic excitations realistic for an inelastic neutron scattering experiments?

2. Following the first question, is there any possibility to make this magnonic excitation appearing at much lower temperature? It will be very interesting if the authors can find a case for the "Weyl magnons" as the lowest excitations in its excitation spectrum.

3. In my mind, the main impact of "Weyl semi-metals" is the transport experimental observation of negative magneto-resistance, which demonstrate "chiral anomaly". I notice that the authors argue the Weyl points in magnonic excitations can be moved in the momentum space by a magnetic field. This means, in my mind, that for "magnonic Weyl point", magnetic field should couples to it as a gauge field. Therefore, one question is if there is any magnonic version of "chiral anomaly" in the current system. Of course, one also needs to worry population effect here for any experimental indication.

4. The authors have shown Berry curvature can exist in magnonic excitation spectrum. This issue has been discussed in two-dimensional magnetic system and early theoretical studies have even shown the possibility of magnonic analog of "Chern insulator" in two-dimensional magnetic materials.

Example includes: Zhang, Lifa, et al. "Topological magnon insulator in insulating ferromagnet." *Physical Review B* 87.14 (2013): 144101.

Mook, Alexander, Jürgen Henk, and Ingrid Mertig. "Edge states in topological magnon insulators." *Physical Review B* 90.2 (2014): 024412.

Shindou, Ryuichi, et al. "Topological chiral magnonic edge mode in a magnonic crystal." *Physical Review B* 87.17 (2013): 174427.

It is well-known that 3D "Weyl semi-metal" with time-reversal breaking is closely related to 2D "Chern insulator". And the Hall conductance in time-reversal breaking "Weyl semi-metal" is directly

related to the distance between two Weyl points in the momentum space. Therefore, here are my questions: (1) are there similar relations between 3D "Weyl magnons" and 2D "Chern insulator" analog of magnonic excitations? (2) Can one get a similar relation between the thermal Hall signal and the distance between Weyl semimetals? Again one needs to worry how to observe these effects due to population effect.

Reviewer #2 (Remarks to the Author):

In this work the authors introduce the notion of "Weyl magnon" which is the magnetic analogue of Weyl fermions in electronic systems. Though Weyl points in quasi-particle spectra have been a focus of considerable interest in current condensed matter physics, their magnetic analogues have not been explored so far in the literature. Taking the breathing pyrochlore antiferromagnet as a specific example, the authors demonstrate that Weyl points do exist in the spin-wave spectrum for a wide range of the parameters. An interesting consequence of Weyl points is the presence of topologically protected arc states localized at the boundaries. The authors then argued that these magnon Weyl points are easier to control than those in electronic systems since the applying magnetic field does not modify the original Brillouin zone. In addition, the authors clarify necessary conditions for the existence of magnon Weyl points and discuss what the experimental implications of them are.

I think the concept of Weyl magnon the authors propose provides a unique opportunity to study topological phenomena in a variety of insulating magnets with nontrivial magnetic orders, and will generate a new stream of research. It should be stressed that the Weyl magnons are conceptually similar to their electronic counterparts, but essentially very different because they couple to the magnetic field only via the Zeeman coupling. This nice property will probably attract much attention from both theorists and experimentalists. In addition, the manuscript is well organized and seems free from detectable errors. Therefore, I would recommend this manuscript for publication in Nature Communications if it is revised to address the following points.

1. Justification of the model

The model Hamiltonian Eq. (1) consists of the standard isotropic exchange and the single-ion anisotropy terms. But from symmetry consideration, the Dzyaloshinskii-Moriya (DM) interaction is also allowed. Since the magnitude of the DM interaction is usually proportional to λ (spin-orbit), while that of the D-term depends quadratically on λ , it is likely that the former is larger than the latter when the spin-orbit is weak. Is there any reason that the DM can be safely neglected here?

2. Figures

Fig.1)

It would be helpful to indicate J and J' bonds in the figure. A short description of phases II, III is also very helpful.

Fig. 2)

This figure is not referred to in the main text. "(b) The magnetic order in region I" should read "(b) The magnetic order in region I and II", because both the regions have the same magnetic structure.

Fig. 4)

E_{Weyl} in the caption is not defined anywhere.

3. Berry curvature

In the discussion about the conditions prohibiting Weyl points, the authors resort to the symmetry property of the Berry curvature. But they do not really illustrate how to compute the Berry

curvature. A procedure to compute it for generic boson systems can be found in Shindou's papers including [Phys. Rev. B 87, 174427 (2013)]. I would suggest that the authors should at least refer to these papers.

4. Thermal Hall effect

One caveat is that a thermal Hall effect may not be a unique signature of the Weyl magnons, because it does occur even in systems without Weyl points. Is there any way to distinguish between the response from Weyl magnons and the other in experiments? If it's so, the authors should touch on this briefly.

Another minor comment is that: in pyrochlore iridates $A_2Ir_2O_7$, a pressure is necessary to break high symmetry so that the anomalous Hall effect takes place. Is there a similar requirement for the magnon Weyl case?

5. surface surface ...

"the surface surface Brioullin zone" in page 5 should be "the surface Brioullin zone".

Response to Reviewers' comments:

Response to Reviewer #1 (Remarks to the Author):

This is an interesting paper about the emergence of topological linear band crossing in the three-dimensional magnonic excitation spectrum, which is dubbed as "Weyl magnons", in analog to "Weyl semi-metals" discovered in electronic systems recently. Based on a physically relevant spin model and the standard spin density wave theory (Holstein-Primarkoff bosons), the authors show that such type of magnonic excitations can be found in the Cr-based breathing pyrochlore lattice. The result is sound and the calculation is standard. In my mind, this paper is possible to arouse research interests of people from the fields of both topological physics and magnetic materials and thus potentially has an impact for both fields. Before I can make a recommendation, I would like to bring the following questions to the authors.

We here thank the referee for the positive comment on our work.

- 1. As discussed in the paper, one worry to observe "Weyl magnons" in experiments is the population effect since the Weyl point in magnonic excitations lies in a relative high energy. This makes the current work less interesting and also difficult for experiments. Could the authors give an estimate of temperature range to observe such type of magnonic excitations in realistic materials? Is the energy range for magnonic excitations realistic for an inelastic neutron scattering experiments?*

Thanks very much for the questions. We would like to explain that even though magnon energies may be high, for example compared to temperature, this does not prevent their measurement. This is because in an inelastic measurement, the energy is supplied not by temperature but externally by for example a neutron. Indeed for such a measurement, an energy scale much larger than temperature is beneficial, since it means that thermal broadening is small compared to the magnon bandwidth.

The transition temperatures (TN) in these breathing pyrochlore systems are 10-15K. If one cools the system below TN, one should certainly be able to observe well-defined spin-wave

excitations. In fact, clear spin-wave spectrum has been observed in the neutron scattering measurement on Yb₂Ti₂O₇ even though the transition temperature is less than 1K in Yb₂Ti₂O₇. Therefore, the magnon excitation can be much higher in energy than the transition temperature.

The energy scale of the magnon excitation is of the order of the exchange J. From the Curie-Weiss temperature, the exchange J would be ~30-50K. This is well in the regime of an inelastic neutron scattering experiment.

2. *Following the first question, is there any possibility to make this magnonic excitation appearing at much lower temperature? It will be very interesting if the authors can find a case for the "Weyl magnons" as the lowest excitations in its excitation spectrum.*

Thanks very much for the remark. First let us reiterate that the magnon can be measured spectroscopically when its energy is large compared to kT, and in fact this is the best regime for such measurements. For this type of measurement there is no reason the Weyl point needs to lie at the energy minimum (and we believe this is not possible). So we do not think this is an important limitation. But see below, as we come more to the thrust of the referee's question.

3. *In my mind, the main impact of "Weyl semi-metals" is the transport experimental observation of negative magneto-resistance, which demonstrate "chiral anomaly". I notice that the authors argue the Weyl points in magnonic excitations can be moved in the momentum space by a magnetic field. This means, in my mind, that for "magnonic Weyl point", magnetic field should couples to it as a gauge field. Therefore, one question is if there is any magnonic version of "chiral anomaly" in the current system. Of course, one also needs to worry population effect here for any experimental indication.*

Thanks very much for the remark. The referee is certainly correct that the negative magneto-resistance of the Weyl semimetal is quite interesting. Yet we would point out that, at least so far, the main impact of Weyl semimetals has not been in transport measurements, but in angle-resolved photoemission (ARPES). In ARPES, one measures states that are not necessarily close to the Fermi energy. This is certainly true in terms of publications. ARPES is an entirely spectroscopic measurement, just as inelastic neutron scattering is for magnons. So in our opinion this is already a sufficient opportunity for impact in experiments.

Yet it is of course interesting to contemplate transport or other effects associated with occupation changes of states near the Weyl point. For magnons, it is not feasible to move this to the chemical potential. However, it may be possible to obtain pseudo-equilibrium states by optical excitation of the corresponding modes. Then one could indeed consider physics like the chiral anomaly. This is certainly not impossible, but checking it out in detail is a subtle problem requiring a thorough analysis of many competing transition rates. We feel this is beyond the scope of the current paper but did in fact comment on the possibility in our manuscript, and think that the question should stimulate future theoretical work.

4. *The authors have shown Berry curvature can exist in magnonic excitation spectrum. This issue has been discussed in two-dimensional magnetic system and early theoretical studies have even shown the possibility of magnonic analog of "Chern insulator" in two-dimensional magnetic materials. Example includes: Zhang, Lifa, et al. "Topological magnon insulator in*

insulating ferromagnet." Physical Review B 87.14 (2013): 144101. Mook, Alexander, Jürgen Henk, and Ingrid Mertig. "Edge states in topological magnon insulators." Physical Review B 90.2 (2014): 024412. Shindou, Ryuichi, et al. "Topological chiral magnonic edge mode in a magnonic crystal." Physical Review B 87.17 (2013): 174427.

Thanks very much for pointing out these references to us. We have added them in the new manuscript.

It is well-known that 3D "Weyl semi-metal" with time-reversal breaking is closely related to 2D "Chern insulator". And the Hall conductance in time-reversal breaking "Weyl semi-metal" is directly related to the distance between two Weyl points in the momentum space. Therefore, here are my questions: (1) are there similar relations between 3D "Weyl magnons" and 2D "Chern insulator" analog of magnonic excitations? (2) Can one get a similar relation between the thermal Hall signal and the distance between Weyl semimetals? Again one needs to worry how to observe these effects due to population effect.

If one fixes one component (say k_z) of the momentum near the magnon Weyl nodes, there exists a gap between the two bands in the k_x - k_y plane, and the lower band carries a non-trivial Chern number. This is an analog of a 2D Chern insulator, as envisioned by the referee. As the referee correctly pointed out, the Hall response of a Weyl semimetal is directly related to the distance between various Weyl nodes in the momentum space. The referee's analogy applies on the thermal Hall transport. The Hall response is proportional to the summation of the Weyl node momenta weighted by their chiralities.

Response to Reviewer #2 (Remarks to the Author):

In this work the authors introduce the notion of "Weyl magnon" which is the magnetic analogue of Weyl fermions in electronic systems. Though Weyl points in quasi-particle spectra have been a focus of considerable interest in current condensed matter physics, their magnetic analogues have not been explored so far in the literature. Taking the breathing pyrochlore antiferromagnet as a specific example, the authors demonstrate that Weyl points do exist in the spin-wave spectrum for a wide range of the parameters. An interesting consequence of Weyl points is the presence of topologically protected arc states localized at the boundaries. The authors then argued that these magnon Weyl points are easier to control than those in electronic systems since the applying magnetic field does not modify the original Brillouin zone. In addition, the authors clarify necessary conditions for the existence of magnon Weyl points and discuss what the experimental implications of them are.

I think the concept of Weyl magnon the authors propose provides a unique opportunity to study topological phenomena in a variety of insulating magnets with nontrivial magnetic orders, and will generate a new stream of research. It should be stressed that the Weyl magnons are conceptually similar to their electronic counterparts, but essentially very different because they couple to the magnetic field only via the Zeeman coupling. This nice property will probably attract much attention from both theorists and experimentalists. In addition, the manuscript is well organized and seems free from detectable errors. Therefore, I would recommend this

manuscript for publication in Nature Communications if it is revised to address the following points.

1. Justification of the model The model Hamiltonian Eq. (1) consists of the standard isotropic exchange and the single-ion anisotropy terms. But from symmetry consideration, the Dzyaloshinskii-Moriya (DM) interaction is also allowed. Since the magnitude of the DM interaction is usually proportional to λ (spin-orbit), while that of the D-term depends quadratically on λ , it is likely that the former is larger than the latter when the spin-orbit is weak. Is there any reason that the DM can be safely neglected here?

Thanks for the question. Actually there is a subtlety behind the perturbative estimate of D-term and DM interaction. If one follows the literature, one has $D \sim (\lambda / \Delta) * \lambda$, where Δ is the crystal field gap. In contrast, the DM interaction would behave as $DM \sim (\lambda / \Delta) * J$. Note λ is $\sim 100\text{cm}^{-1} \sim 120\text{K}$ for late transition metal ions while the exchange J can be estimated from Curie Weiss temperature and is $\sim 30\text{-}50\text{K}$ in the experimental refs. Moreover, even after a small DM interaction is included, because the Weyl node is topologically protected and robust, the small DM interaction cannot immediately remove the Weyl nodes.

2. Figures

Fig.1) It would be helpful to indicate J and J' bonds in the figure. A short description of phases II, III is also very helpful.

Thanks for pointing it out. We have modified that in the new manuscript.

Fig. 2) This figure is not referred to in the main text. "(b) The magnetic order in region I" should read "(b) The magnetic order in region I and II", because both the regions have the same magnetic structure.

Thanks for pointing it out. We have modified that in the new manuscript.

Fig. 4) E_{Weyl} in the caption is not defined anywhere.

Thanks again for pointing it out. We have fixed it in the new manuscript.

3. Berry curvature In the discussion about the conditions prohibiting Weyl points, the authors resort to the symmetry property of the Berry curvature. But they do not really illustrate how to compute the Berry curvature. A procedure to compute it for generic boson systems can be found in Shindou's papers including [Phys. Rev. B 87, 174427 (2013)]. I would suggest that the authors should at least refer to these papers.

Thanks for suggesting this reference. We have added it.

4. Thermal Hall effect One caveat is that a thermal Hall effect may not be a unique signature of the Weyl magnons, because it does occur even in systems without Weyl points. Is there any way to distinguish between the response from Weyl magnons and the other in experiments? If it's so, the authors should touch on this briefly. Another minor comment is that: in pyrochlore iridates $A_2\text{Ir}_2\text{O}_7$, a pressure is necessary to break high symmetry so that the anomalous Hall effect takes place. Is there a similar requirement for the magnon Weyl case?

We agree with the referee that thermal Hall effect may not be a unique signature of Weyl magnon. But it is an important consequence of the Weyl magnon. The direct confirmation would be the observation of Weyl nodes and surface states.

In pyrochlore iridates, the all-in-all-out magnetic order preserves the cubic symmetry, the contribution from the Weyl nodes cancels with each other. That is because the Hall conductance is proportional to the summation of the Weyl node momentum weighted by their chiralities. If cubic symmetry is preserved, the Hall response will just vanish. Similar behavior occurs for Weyl magnon. However, here we can strongly break the cubic symmetry with an applied field.

5. surface surface ... "the surface surface Brillouin zone" in page 5 should be "the surface Brillouin zone".

We are very impressed with your care. Thanks.

REVIEWERS' COMMENTS:

Reviewer #1 (Remarks to the Author):

The authors have answered my questions satisfactorily and improved the manuscript accordingly. I think the current version of the paper is ready for the publication in Nature Communications.

Reviewer #2 (Remarks to the Author):

I have carefully examined the revised manuscript and the response to Reviewers' comments. I am satisfied with their replies to my comments and the changes the authors have made. In addition, the concerns raised by the other Reviewers have also been addressed satisfactorily. Therefore, I recommend this manuscript for publication in Nature Communications without further modifications.

RESPONSE TO REVIEWERS' COMMENTS:

RESPONSE To Reviewer #1 :

The authors have answered my questions satisfactorily and improved the manuscript accordingly. I think the current version of the paper is ready for the publication in Nature Communications.

Thanks very much for the recommendation.

RESPONSE To Reviewer #2 :

I have carefully examined the revised manuscript and the response to Reviewers' comments. I am satisfied with their replies to my comments and the changes the authors have made. In addition, the concerns raised by the other Reviewers have also been addressed satisfactorily. Therefore, I recommend this manuscript for publication in Nature Communications without further modifications.

Thanks very much for being very careful again and also for recommendation.